

# Portable Ozone Calibration Source Independent of Changes in Temperature, Pressure and Humidity for Research and Regulatory Applications

John W. Birks, Craig J. Williford, Peter C. Andersen, Andrew A. Turnipseed, Stanley Strunk, and Christine
A. Ennis

2B Technologies, Inc., 2100 Central Ave., Boulder, CO 80301
*Correspondence to:* John Birks (johnb@twobtech.com)

**Abstract:**

A highly portable ozone ($O_3$) calibration source that can serve as a U.S. EPA Level 4 transfer standard for
the calibration of ozone analyzers is described and evaluated with respect to analytical figures of merit
and effects of ambient pressure and humidity.  Reproducible mixing ratios of ozone are produced by the
photolysis of oxygen in $O_3$-scrubbed ambient air by UV light at 184.9 nm light from a low pressure
mercury lamp.  By maintaining a constant volumetric flow rate (thus constant residence time within the
photolysis chamber), the mixing ratio produced is independent of both pressure and temperature and
can be varied by varying the lamp intensity.  Pulse width modulation of the lamp with feedback from a
photodiode monitoring the 253.7-nm emission line is used to maintain target ozone mixing ratios in the
range 30-1,000 ppb.  In order to provide a constant ratio of intensities at 253.7 and 184.9 nm, the
photolysis chamber containing the lamp is regulated at a temperature of 40 °C.   The resulting $O_3$
calibrator has a response time for step changes in output ozone mixing ratio of < 20 s and precision ($\sigma_p$)
of 0.4% of the output mixing ratio for 10-s measurements (e.g., $\sigma_p$ = ± 0.4 ppb for 100 ppb of $O_3$).
Ambient humidity was found to affect the output mixing ratio of ozone primarily by dilution of the
oxygen precursor.  This potential humidity interference could be up to a few percent in extreme cases
but is effectively removed by varying the lamp intensity to compensate for the reduced oxygen
concentration based on feedback from a humidity sensor.



## 1      Introduction


Ozone ($O_3$) is a key constituent throughout the atmosphere.  In the lower atmosphere, it is a secondary air pollutant formed by the interaction of sunlight with primary pollutants consisting of oxides of nitrogen, carbon monoxide and volatile organic compounds (e.g., Haagen-Smit, 1952; Birks, 1998; Sillman, 1999).  Because of its adverse health effects, ozone is one of six Criteria Pollutants designated

by the U.S. Environmental Protection Agency (U.S.-EPA, 2018).  Although ground-level ambient ozone levels have improved over the past few decades, many regions in the U.S. are still out of compliance with the National Ambient Air Quality Standard (NAAQS) for ozone, and monitoring of ozone at hundreds of State and Local Air Monitoring Sites (SLAMS) is mandated by the EPA.

In the stratosphere ozone is continuously formed in the photolysis of oxygen by UV light having

wavelengths less than 242 nm.  The ozone produced absorbs UV light, protecting the Earth from harmful UV-B radiation in the wavelength range 280-320 nm.  Monitoring of the protective ozone layer is done by use of ground-based spectroscopic methods (Gotz, et al., 1934; Stone et al., 2015) along with balloon-launched ozonesondes (Komhyr, 1969), occasional aircraft measurements, and satellites.

Ozone has also long been used industrially for treatment of drinking water (Guinvarch, 1959;

Lebout, 1959; Peleg, 1976; Rice, 1996), and there is a rapidly growing number of other applications involving food processing, deodorization, sanitization and sterilization (e.g., Jordan and Carlson, 1913; Kim, 1999; Karaca and Velioglu, 2007).  As a result, ozone measurements are required for monitoring industrial processes and insuring the health and safety of workers.

All of these areas of study require monitoring of ozone levels in either air or water.  Although

there are numerous methods for measuring ozone, the UV absorbance technique at the 253.7-nm emission line of a low pressure mercury lamp is now almost universally used.  Absorbance has the advantage of being an "absolute" method (in theory relying only on the optical pathlength and absorption cross section of the analyte); however, UV photometers used to measure ozone do still require periodic calibration.  Since environmental ozone-monitoring applications often require relatively

long-term, continuous measurements, systematic errors can arise due to drift of electrical components (e.g., A/D converters, temperature and pressure sensors) or degradation of instrument components such as the sampling pump or $O_3$ scrubber.  Errors due to incomplete flushing of the detection cell between analyte and reference measurements of light intensity can result from reduced pumping efficiency.  Incomplete scrubbing of ozone during the reference light intensity measurement, as well as

adsorption/desorption of UV-absorbing species such as aromatic VOCS and elemental mercury from the ozone scrubber (Spicer et al., 2010; Turnipseed et al, 2017), and the effects of changing humidity levels





on light transmission through the detection cell (Wilson and Birks, 2006) can all affect the photometer's linearity and offset. Ozone instruments based on other techniques such electrochemical ozonesondes (Komhyr, 1969) or solid-phase or gas-phase chemiluminescence (Regener, 1964; Güsten et al., 1992)

also are known to be sensitive to many variables that can induce systematic errors and often require even more frequent calibration checks. As a result, periodic calibrations of ozone monitors of all types are required, and a portable calibrator is highly desirable, especially for instruments deployed in remote locations.

Because ozone is an unstable gas, easily decomposing to molecular oxygen, calibrations require

generating ozone at known concentrations at the site of the ozone monitor to be calibrated. This is done almost universally by use of an ozone calibration source in which ozone is generated by photolysis of $O_2$ at 184.9 nm using a low pressure mercury lamp. Most commonly, the calibrator dries the ambient air or uses dry air from a compressed gas cylinder to eliminate biases due to water vapor and incorporates an ozone photometer that continuously measures the ozone produced. The target output

mixing ratio of ozone is then controlled in a feedback loop that regulates the lamp intensity. Such calibrators are relatively large, heavy and have high power requirements. A more portable instrument such as the one described here can regulate ozone output mixing ratios solely based on feedback from measurements of the lamp intensity and does not require dry air or a built-in photometer.

For regulatory purposes, ozone measurements must be traceable to a fundamental reference

standard. In the U.S., the EPA originally prescribed a wet chemical technique for ozone calibrations based on the spectrophotometric analysis of iodine generated by $O_3$ in a neutral potassium iodide solution (NBKI method) that itself was referenced to an arsenious oxide primary standard (Beard et al., 1977). That method was replaced in 1979 with direct absorbance in the gas phase, now using an accepted value for the absorption cross section for $O_3$ at 253.7 nm of $1.15 \times 10^{-17}$ cm$^2$ molec$^{-1}$

(Burkholder et al., 2015). The U.S. and many other nations are members of the Convention of Meter, which makes use of the International Bureau of Weights and Measures (BIPM) Standard Reference Photometer #27 as the world's ozone reference standard (Paur et al., 2003). Each member state of the Convention of the Meter has one laboratory designated to provide traceability to that country. For the U.S. that laboratory is the National Institute for Standards and Technology (NIST). Standard Reference

Photometers (SRPs) are maintained by both NIST and the EPA. The calibrations of regulatory ozone monitors in the U.S. are traceable to these Level 1 SRPs via transfer standards, as detailed in Fig. 1. This figure also shows how EPA-maintained SRPs trace back through the NIST Standard Reference Photometer #0 (SRP#0) to the world standard, SRP #27. Once every two years, the NIST SRP #2 is



calibrated against the NIST SRP #0.  The EPA Office of Research and Development Metrology maintains

EPA SRP #1 and #7, and these are verified against the NIST SRP #2 once each year.  Verification requires

that a linear regression of the photometer ozone output plotted against the NIST SRP have a slope of

1.00±0.01 and intercept of ±1 ppb; i.e., 1% agreement.  Upon verification, EPA SRP #7 is sent to the

different EPA regions for verification of their respective SRPs.  As further verification, EPA SRP #7 is

occasionally compared to EPA SRP #1.

Transfer standards are defined as "a transportable device or apparatus which, together with

associated operation procedures, is capable of accurately reproducing pollutant concentration

standards or produce accurate assays of pollutant concentrations which are quantitatively related to a

higher level and more authoritative standard" (U.S.-EPA, 2013).  Thus, a transfer standard for ozone can

be either an ozone source or an ozone analyzer.  The EPA accepts up to four levels of ozone transfer

standards for calibration of an ozone monitoring site or field ozone analyzer, as shown in Fig. 1.  Also, as

illustrated in this figure, the uncertainty increases with each level of transfer standard.  Typically, a Level

2 "uncompromised standard" is maintained in the laboratory where conditions of use may be carefully

controlled.  This transfer standard is used to calibrate Level 3 transfer standards that encounter frequent

use and potentially rough treatment in the field.  The Level 3 transfer standards may be returned on a

frequent basis for verification by the Level 2 standard.  Level 4 standards, calibrated against Level 3

standards, also are allowed.  Often, level 3 and 4 standards are more portable and designed to be more

rugged and/or less sensitive to environmental conditions than higher level transfer standards.  They may

be used for calibrating instruments deployed in remote locations, for example.

An EPA Level 2 transfer standard must include both an ozone generation device and an analyzer.

A Level 3 transfer standard can be a combination of an ozone generator and analyzer or only an

analyzer.  A Level 4 transfer standard can be an ozone analyzer or only an ozone generation device.

Thus, the ozone calibration source described here qualifies as a Level 4 transfer standard.  Levels 2-4

Transfer Standards must undergo a "6x6" verification in which six calibration curves, each consisting of

six approximately equally spaced ozone concentrations in a range including 0 and 90% (±5%) of the

upper range of the reference standard, is obtained on six different days (U.S.-EPA, 2013).  The relative

standard deviations of the six slopes of the calibration plots must not exceed 3.7%, and the standard

deviation of the 6 intercepts cannot exceed 1.5 ppb.

Here we describe a portable, low-cost ozone calibrator that meets the specifications as an EPA

Level 4 transfer standard. The calibrator is low power, requiring only 18 watts of power, and does not

require the inlet air to be dried.  It is independent of both temperature and pressure and corrections





due to humidity are easily incorporated.  Therefore, it can provide accurate and precise ozone mixing ratios for calibration of field analyzers or can be used as a reliable ozone source in laboratory experiments.

**2      Experimental**

The 2B Technologies Model 306 Ozone Calibration Source™ described here makes use of a low pressure mercury (Hg) lamp to photolyze oxygen in ambient air to produce known mixing ratios of ozone.  The vacuum UV lines at 184.9 nm are absorbed by $O_2$ to produce oxygen atoms.  The oxygen atoms rapidly attach to $O_2$ to form ozone molecules according to the same mechanism that is

responsible for the presence of Earth's protective ozone layer:

$$O_2 + h\nu \rightarrow O + O \qquad (1)$$

$$2 [O + O_2 + M \rightarrow O_3 + M] \qquad (2)$$

_______________________________

$$\text{Net: } 3 O_2 + h\nu \rightarrow 2 O_3 \qquad (3)$$

where $h\nu$ symbolizes a photon of light and M is any molecule (e.g., $N_2$, $O_2$, Ar).  Absorption of one photon of 184.9-nm light by $O_2$ results in the formation of two ozone molecules.  The concentration of ozone produced in a flowing stream of air depends on the intensity of the photolysis lamp, the concentration of oxygen (determined by pressure, temperature and its mixing ratio in air), and the residence time in the photolysis cell (determined by volumetric flow rate and cell volume).  As will be

discussed below, pressure and temperature affect the concentration of the ozone produced (e.g.,  molec $cm^{-3}$), but do not affect the output mixing ratio (e.g., ppb).  Thus, by holding the volumetric flow rate constant, it is possible to produce a flow of air containing a constant mixing ratio of ozone that can be varied most conveniently by changing and controlling the lamp intensity.

Figure 2 is a schematic diagram of the 2B Technologies Model 306 Ozone Calibration Source.

Ambient air is forced by an air pump through a particulate filter, a mass flow meter, and a chemical scrubber to remove ozone and $NO_x$ (= NO + $NO_2$), before entering the photolysis chamber containing a low-pressure mercury lamp where absorption of 184.9-nm photons by oxygen produces ozone.  The lamp intensity at 253.7 nm is monitored by a photodiode having a built-in interference filter centered at 254 nm and is controlled by the microprocessor in a feedback loop to maintain a target output ozone

mixing ratio.  Note that the lamp emission at 253.7 nm, which is not absorbed by oxygen to make ozone, is monitored instead of the 184.9 nm line.  This is because the window that separates the photodiode from the photolysis chamber is much more susceptible to changes in transmission due to deposition of





UV-absorbing materials at 184.9 nm than at 253.7 nm. In order to maintain a constant ratio of emission

intensities of the Hg lamp at 184.9 and 253.7 nm, the photolysis chamber temperature is regulated at 40

°C by means of a temperature sensor and heating cartridge. Pressure within the gas stream is measured

but not controlled. The residence time is held constant by ensuring a constant volumetric flow rate

using a mass flow meter (TSI Instruments, Model 4040) converted to volumetric flow using the

measured temperature and pressure of the photolysis cell. A microprocessor reads the output of the

mass flowmeter, temperature and pressure of the photolysis chamber, and regulates the volumetric

flow rate be 3.0 L min$^{-1}$ by means of pulse-width modulation of the power supplied to the pump. In

addition to controlling the volumetric flow rate the target photodiode voltage is scaled to the

instantaneously measured volumetric flow rate in order to compensate for flow rate fluctuations, (e.g.,

higher flow rates require higher target photodiode voltages).

Air containing ozone exits the photolysis cell through an overflow tee, where excess air that is

not drawn by the ozone monitor being calibrated is exhausted through an internal ozone scrubber. The

output of the ozone calibration source may be attached directly to any ozone monitor (providing that its

sampling rate is less than 3.0 L min$^{-1}$); excess ozone flow is diverted through the ozone scrubber internal

to the calibrator, and any perturbation in total flow rate is automatically adjusted by the microprocessor

using feedback from the mass flow meter. A three-way solenoid valve is installed just before the exit of

the calibrator that allows the ozone calibration source to be plumbed in-line with the sampling inlet to

an ozone monitor, so that the monitor can sample either ambient air or the output of the calibrator.

The output of the ozone source is calibrated using a reference ozone monitor with traceability to NIST,

and slope and offset calibration parameters are determined from linear regression and applied to the

target photodiode voltages to achieve target ozone mixing ratios.


### 3    Results and Discussion

### 3.1    Linearity, Reproducibility and Precision of Output Concentration

An example of stepwise outputs of a Model 306 Ozone Calibration Source is provided in Fig. 3.

The target output ozone mixing ratio was varied in the range of 0 to 1,000 in steps of 0, 50, 100, 200,

400, 600, 800 and 1,000 ppb. This was followed by a series of decreasing steps back to 0 ppb. A second

set of stepwise increases and decreases in target ozone concentrations followed. Each step

concentration was maintained for ~5 minutes (30 measurements). Output ozone concentrations were

measured and logged every 10 s by a 2B Technologies Model 202 Ozone Monitor, a U.S. EPA Federal

Equivalent Method (FEM). Note that the response time to achieve a new target concentration is 3 or



fewer data points (< 30 s). The response of the calibration source is actually faster considering that it is

convolved with the Model 202 Ozone Monitor which outputs the average of the most recent two 10-s

measurements. Figure 4 is a plot of average measured ozone concentration vs target concentration for

the data of Fig. 3. Linear regression lines are drawn for the two stepwise increases and two stepwise

decreases in target ozone concentration. The data points and four regression lines overlap so well that

they cannot be distinguished on the graph. The equations for the linear regression lines have slopes

that agree to better than ±1%, and the standard deviation of the four intercepts is 1.3 ppb. The

coefficients of determination ($R^2$) are all 0.9999 or 1.0000.

The precisions ($1\sigma_p$) of the measured output ozone mixing ratios vary from 2.1 ppb at 0 ppb

ozone (i.e., the measurement precision of the Model 202 ozone monitor) to 6.2 ppb at 1,000 ppb ozone.

A plot of precision vs ozone concentration (data not shown) gives a straight line with intercept of 1.8

ppb, slope of 0.0042 ppb/ppb $O_3$ and $R^2$ of 0.9586. Thus, the precision of the ozone output is about

0.4% of the target concentration (e.g., ±0.4 ppb at 100 ppb $O_3$ and ±4 ppb at 1,000 ppb $O_3$).

In order to verify the ability of the Model 306 Ozone Calibration Source to qualify as a US EPA

Level 4 Transfer Standard (US EPA, 2013), we carried out a "6x6" calibration in which we measured the

output of the ozone calibration source at six different target ozone concentrations (50, 100, 150, 200,

250, and 300 ppb) in addition to a zero ozone measurement on six consecutive days. The ozone output

mixing ratios were measured using a 2B Technologies Model 205 FEM ozone monitor. As can be seen in

Table 1, the instrument easily met the requirements (given in Table 3-1 of US EPA, 2010) of a Level 4

standard with a measured relative standard deviation (RSD) of 0.26% for the slopes of the regression

plots vs. the requirement of $\leq 3.7\%$ and a measured standard deviation of 0.33 ppb of the intercepts vs.

the requirement of $\leq 1.5$ ppb. Values for the coefficient of determination ($R^2$) were in the range of

0.9998 to 1.0000 with an average of 0.9999 for the six calibration plots.

Other specifications of that are of interest for portability (such as the size, weight and power

requirements) are given in Table 2.


### 3.2    Effect of Pressure on the Ozone Output Mixing Ratio

As described earlier, the target mixing ratio output of the ozone calibration source is achieved

by varying the photolysis lamp intensity and maintaining a constant volumetric flow rate. Pressure

within the gas stream is measured to correct the mass flow measurements, but not controlled, since the

goal is to produce a constant mixing ratio (mole fraction) of ozone rather than a constant concentration.

The absorption cross section ($\sigma_{O2}$) for $O_2$ at the 184.9 nm Hg line is still poorly known due to significant





fine structure in the spectrum but is approximately 1 x 10⁻²⁰ cm² molec⁻¹ (Yoshino et al., 1997), and the oxygen concentration ($c_{O2}$) in air at a temperature of 298 K and pressure of 1 atm is 5.2 x 10¹⁸ molec cm⁻³. The average path length ($l$) of the ozone calibration source was designed to be ~0.25 cm, making

the absorbance ($\sigma_{O2}lc$) optically thin with a single path absorbance of ~1.3 x 10⁻³; i.e., only 0.13% of the 184.9-nm light emitted by the lamp is absorbed by oxygen.  Under optically thin conditions, the ozone production rate ($P_{O3}$) within the photolysis chamber is given by

$$P_{O3} = 2I\sigma_{O2}c_{O2} = 2I\sigma_{O2}(0.2095 c_{air}) \tag{4}$$


where $I$ is the lamp intensity (photons cm⁻² s⁻¹) at 184.9 nm, and $c_{O2}$ is the concentration of oxygen molecules (molec cm⁻³), which make up 20.95% of dry air.  The factor of 2 accounts for the production of two ozone molecules for every oxygen molecule photolyzed.  The output mixing ratio of ozone (fraction of air molecules that are ozone), $X_{O_3}$, in ppb is then given by


$$X_{O_3}(ppb) = \frac{\left(P_{O3}, \frac{molec}{cm^3 s}\right)(\tau_{cell}, \ s)}{\left(c_{air}, \frac{molec}{cm^3}\right)} \times 10^9 = \frac{2I\sigma_{O2}(0.2095)V}{F} \times 10^9 \tag{5}$$

where $\tau_{cell}$ is the residence time of the photolysis cell, which is equal to the cell volume ($V$) divided by the volumetric flow rate, $F$, and $P_{O3}$ is given by equation 4.   Note that the total molecular concentration of

air in the denominator of equation 5 cancels with the air concentration in the numerator, so the ozone mixing ratio output is independent of molecular concentration and therefore independent of chamber pressure and temperature (although chamber temperature is controlled for a separate reason described in Section 2).  The only parameters that affect the ozone output mixing ratio are the lamp intensity and volumetric flow rate.  As mentioned before, the volumetric flow rate is computed from the measured

mass flow rate, temperature, and pressure, and is maintained at 3 L min⁻¹.

In order to test for the predicted independence of ambient pressure, the output of a calibrated Model 306 Ozone Calibration Source was measured at six programmed ozone concentrations (0, 100, 200, 300, 400, and 500 ppb) in Boulder, Colorado (5,430 ft, 1,655 m altitude; P ≅ 0.82 atm) and at Fritz Peak (9,020 ft, 2,749 m altitude; P ≅ 0.71 atm) in the mountains west of Boulder.  The results are shown

in Fig. 5. The output ozone mixing ratios are at these two altitudes are indistinguishable, as predicted by theory.



### 3.3 Effect of Water Vapor on the Ozone Output Mixing Ratio

Water vapor could potentially affect the output ozone concentration in several ways. The first is simply by dilution. As the relative humidity increases, the partial pressure and therefore molecular concentration of $O_2$ decreases, resulting in a reduced production rate of ozone. The water vapor mixing ratio in the atmosphere is typically ~2% by volume but could be as high as 7.3% (100% RH at 40 °C), resulting in a 7.3% reduction in ozone output in highly humid air if the ozone calibration source were originally calibrated in dry air.

Another way that water vapor can reduce the output ozone mixing ratio is by attenuating the lamp intensity through absorbance. The absorption cross section for $H_2O$ at 184.9 nm is $7.14 \times 10^{-20}$ $cm^2$ $molec^{-1}$ (Cantrell et al., 1997). In the extreme case mentioned above of a water vapor mixing ratio of 7.3% ($[H_2O] = 1.8 \times 10^{18}$ molec $cm^{-3}$), the average fraction of 184.9-nm light absorbed by water vapor at atmospheric pressure and 40° C integrated over the 0.25 cm path length is 1.6%. An offsetting factor is that the mass flow controller is 15.4% more sensitive to water vapor ($C_p = 33.59$ J $K^{-1}$ $mol^{-1}$) than to air ($C_p = 29.10$ J $K^{-1}$ $mol^{-1}$) due to its higher heat capacity (NIST, 2018). Increasing the water vapor mixing ratio results in a positive error in the measured flow rate, with the result that the air pump is slowed down in the feedback loop to maintain a constant apparent flow rate and the residence time in the photolysis cell is increased. For a 7.3% increase in water vapor, this effect results in a 1.1% increase in ozone output. Thus, these two factors – the attenuation of 184.9-nm light by water vapor and the reduced flow rate due to change in heat capacity of the sample air – offset one another to within ~0.5% in expected ozone output.

Yet another way that humidity could affect ozone production is through secondary photochemical reactions. The photochemistry of water vapor is rather complicated, especially in the presence of ozone. $HO_x$ radicals (OH and $HO_2$) are produced directly by photolysis of water vapor,

$$H_2O + h\nu \rightarrow OH + H \quad\quad\quad (6)$$

$$H + O_2 + M \rightarrow HO_2 + M \quad\quad\quad (7)$$

and indirectly in the reaction of $O(^1D_2)$ with water vapor. $O(^1D_2)$ is produced in the photolysis of ozone at the principal mercury line of 253.7 nm where ozone has a strong absorption,

$$O_3 + h\nu \rightarrow O_2 + O(^1D_2) \qu\quad\quad\quad (8)$$

Although most of the $O(^1D_2)$ is quenched by oxygen and nitrogen in the air stream, a small fraction can react with water, producing OH,

$$O(^1D_2) + H_2O \rightarrow 2\ OH \qu\quad\quad\quad (9)$$



Hydroxyl radicals participate in a well-known, yet relatively slow, catalytic cycle for ozone destruction

(Bates and Nicolet, 1950):

$$OH + O_3 \rightarrow HO_2 + O_2 \tag{10}$$

$$HO_2 + O_3 \rightarrow OH + 2\,O_2 \tag{11}$$

$$\overline{\hspace{3cm}}$$

$$\text{Net: } 2\,O_3 \rightarrow 3\,O_2 \tag{12}$$

But the concentration of hydroxyl radicals that build up inside the photolysis chamber is limited by its

self-reaction, which actually produces ozone,

$$OH + OH \rightarrow H_2O + O \tag{13}$$

$$O + O_2 + M \rightarrow O_3 + M \tag{2}$$

and by the very fast chain termination reaction of OH and $HO_2$:

$$OH + HO_2 \rightarrow H_2O + O_2 \tag{14}$$

Reaction (14) limits the importance of the self-reaction of $HO_2$,

$$HO_2 + HO_2 \rightarrow H_2O_2 + O_2 \tag{15}$$

which also serves to remove $HO_2$. Subsequent photolysis of the $H_2O_2$ product could regenerate OH, but

this was found to have no significant effect on the output mixing ratio of ozone in the model calculations

discussed below, likely due to the low amounts of $H_2O_2$ produced.

The photochemistry within the photolysis chamber was modeled using current

recommendations for the absorption cross sections and reaction rate constants of relevant reactions

summarized in Table 3.  Light intensity at 184.9 nm was adjusted in the model to produce desired output

mixing ratios of ozone in the range 0-1,000 ppb in the absence of water vapor.  Model results for a

target output concentration of 100 ppb ozone are summarized in Fig. 6.  In the extreme case of a

temperature of 40 °C and 100% RH (water mixing ratio of 7.3%), the ozone output mixing ratio increases

by 0.9% (0.9 ppb) due to production of O atoms in the OH self-reaction, reaction 13.  For more typical

conditions of 25 °C and 50% RH, the increase in ozone production is only 0.2% for a target mixing ratio

of 100 ppb.  For a target of 1,000 ppb, the percentage increase in ozone production is slightly smaller,

being only 0.06% (0.6 ppb) at 40 °C and 100% RH.  Under these conditions the catalytic ozone

destruction cycle of reactions 9 and 10 begin to offset ozone production in the OH self-reaction.  Under

more typical conditions of 25 °C and 50% RH, the increase in ozone concentration is modeled to be less

than 0.01% (less than 0.1 ppb) for a target of 1,000 ppb ozone.

Based on the analysis given above, the only significant effect of water vapor (> 1%) on the

output of the ozone calibration source is the dilution of oxygen in the inlet air.  In order to correct for





the dilution effect, a humidity sensor (Honeywell, HIH8000) was installed in the flow path immediately

upstream of the photolysis cell, and feedback from that sensor was used to adjust the lamp intensity to

compensate for dilution of oxygen by water vapor.  The sensor provides simultaneous measurements

relative humidity (RH) and temperature so that mixing ratios of water vapor may be calculated.  Several

empirical equations have been developed to fit the vapor pressure of water as a function of RH and

temperature.  The Magnus-Tetens equation (Tetons, 1930; Montieth and Unsworth, 2008) is sufficiently

accurate while being simple:

$$P_{H2O}(mbar) = 6.1078 \exp\left(\frac{17.27*T(°C)}{T(°C)+237.3}\right) \qquad (16)$$


The mixing ratio of water is then given by:

$$X_{H2O} = \frac{P_{H2O}(mbar)}{P(total)} \times \%RH/100 \qquad (17)$$

Water dilutes the oxygen in the photolysis chamber and therefore reduces the output of the ozone

source by the same factor.  In order to compensate, we may increase the lamp target intensity by a

factor of $1/(1 - X_{H2O})$, and the overall factor we need to multiply the target lamp intensity by is:

$$\frac{1}{\left[1-\frac{6.1078}{P_{Total}}exp\left(\frac{17.27*T(°C)}{T(°C)+237.3}\right)\frac{\%RH}{100}\right]} \qquad (18)$$


In order to test this algorithm, we measured the output of a 2B Technologies Model 306 Ozone

Calibration Source with and without water vapor added.  A three-way valve directed a volumetric flow

rate of 3 L min$^{-1}$ of dry zero air (US Welding) from a compressed gas cylinder to either bypass or pass

through a Nafion® tube immersed in a temperature-controlled water bath to provide either dry air or

humidified air to the inlet of the Model 306.  The output of the ozone calibration source was sampled by

a 2B Technologies Model 211 Ozone Monitor, which because of its gas-phase-scrubber technology and

internal DewLine™ (Nafion® tube) to equilibrate humidity levels of ozone scrubbed and unscrubbed air,

has no significant sensitivity to water vapor.  Experiments were performed with and without lamp

intensity adjustment controlled by the instrument firmware to correct the presence of water vapor.

Figure 7a shows the calibration curves obtained for ozone in the range 0-200 ppb at 0% RH (bypass) and





an average of 82% RH (water vapor added via Nafion® tube) under ambient conditions of 875 mbar pressure and temperature of 23.6 °C and with no lamp intensity adjustment for humidity. The slope of the regression line in the presence of humidity is 2.8% lower than that for dry air, which agrees extremely well with the mixing ratio of water calculated to be 2.7%. Figure 7b shows the calibration

curves obtained for zero air and for humid air (90% RH at 23.8 °C, 3.2% water vapor) where the calibrator lamp intensity is corrected for the dilution due to humidity. As seen in the figure, the slopes are now within 0.1% of each other (0.9929 for dry air and 0.9917 for humid air, i.e., no statistical difference).

**4      Conclusions**

The 2B Technologies Model 306 Ozone Calibration Source is capable of producing ozone in ozone-scrubbed ambient air with accuracy and precision better than 1 ppb in the range 30-100 ppb ozone or 1% in the range 100-1,000 ppb. The volumetric flow rate of 3 L min$^{-1}$ allows calibration of virtually any ozone monitor via sampling from a built-in overflow tee. The instrument is made

independent of ambient pressure and temperature by feedback control of the air pump to produce a constant volumetric flow rate through the photolysis chamber. Regulation of the photolysis chamber temperature, typically at 40 °C, assures a constant ratio of lamp intensities at 184.9 nm (used to photolyze $O_2$) and 253.7 nm (monitored for feedback control of the lamp intensity). The effect of ambient humidity on ozone production is primarily that of dilution of the $O_2$ photochemical precursor.

This dilution effect is completely eliminated by means of feedback control of the photolysis source intensity based on real time measurements of humidity. Photochemical reactions involving $HO_x$ species due to the presence of water vapor only contribute to ozone production by a small amount (< 1% at 40 °C and 100% RH). The ozone calibration source described here is low power (~ 18 W) and highly portable, weighing only 2.6 kg and requiring no compressed or dry gas sources. Yet it still meets the

requirements of an EPA Level 4 transfer standard that can be used in the calibration of compliance-monitoring ozone monitors.

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





Table 1. Results of a US EPA "6x6" calibration of the Model 306 Ozone Calibration Source.


| Day | Slope | Offset, ppb | $R^2$ |
|---|---|---|---|
| 1 | 1.0031 | 0.37 | 0.9998 |
| 2 | 1.0032 | -0.22 | 0.9998 |
| 3 | 1.0054 | -0.05 | 0.9999 |
| 4 | 1.0088 | -0.47 | 0.9999 |
| 5 | 1.0072 | 0.29 | 0.9999 |
| 6 | 1.0021 | 0.21 | 1.0000 |
| **Average** | 1.0050 | 0.02 | 0.9999 |
| **Std. Dev.** | 0.0026 | 0.33 | 0.0001 |



Table 2. Analytical and Physical Specifications for Ozone Calibration Source


| Method of Ozone Production | UV Photolysis of $O_2$ at 184.9 nm |
|---|---|
| Output Concentration Range | 0 ppb and 30 to 1,000 ppb |
| Precision and Accuracy of Output | Greater of 2 ppb or 2% of ozone concentration[1] |
| Response Time for Change in Ozone Output Concentration | 30 s to reach 95% of concentration change |
| Output Flow Rate | 3.0 Liter min$^{-1}$ volumetric |
| Power Requirements | 12 V dc or 120/240 V ac, 18 watt |
| Size | 3.5 x 8.5 x 11 in (9 x 21 x29 cm) |
| Weight | 5.6 lb (2.6 kg) |

[1]The 2B Technologies specification for precision and accuracy of the Model 306 Ozone Calibration Source given here is larger than found in this work and accounts for potential variability among individual instruments.






Table 3. Thermal and photochemical reactions used in modeling the effects of water vapor on the output of the ozone calibration source at 40 °C and 1 atm.  Units are $cm^2$ $molec^{-1}$ for absorption cross sections, $cm^3$ $molec^{-1}$ $s^{-1}$ for second order reactions and $cm^6$ $molec^{-2}$ $s^{-1}$ for third order reactions.

| Reaction | Rate Coefficient or Absorption Cross Section | Reference |
|---|---|---|
| $O_2 + h\nu$ (184.9 nm) $\rightarrow$ 2 O $\rightarrow$ 2 $O_3$ | $1.0 \times 10^{-20}$ | Yoshino et al, 1992 |
| $H_2O + h\nu$ (184.9 nm) $\rightarrow$ OH + H $\rightarrow$ OH + $HO_2$ | $7.14 \times 10^{-20}$ | Cantrell et al., 1997 |
| $O_3 + h\nu$ (253.7 nm) $\rightarrow$ $O_2$ + O($^1D_2$) | $1.15 \times 10^{-17}$ | Burkholder et al., 2015 |
| OH + $HO_2$ $\rightarrow$ $H_2O$ + $O_2$ | $1.01 \times 10^{-10}$ | Burkholder et al., 2015 |
| OH + $O_3$ $\rightarrow$ $HO_2$ + $O_2$ | $8.45 \times 10^{-14}$ | Burkholder et al., 2015 |
| $HO_2$ + $O_3$ $\rightarrow$ OH + 2 $O_2$ | $2.09 \times 10^{-15}$ | Burkholder et al., 2015 |
| OH + OH $\rightarrow$ $H_2O$ + O $\rightarrow$ $H_2O$ + $O_3$ | $1.8 \times 10^{-12}$ | Burkholder et al., 2015 |
| OH + OH (+M) $\rightarrow$ $H_2O_2$ (+M) | $1.59 \times 10^{-11}$ | Burkholder et al., 2015 |
| $HO_2$ + $HO_2$ $\rightarrow$ $H_2O_2$ + $O_2$ | $1.30 \times 10^{-12}$ | Burkholder et al., 2015 |
| $HO_2$ + $HO_2$ + M $\rightarrow$ $H_2O_2$ + $O_2$ | $3.96 \times 10^{-32}$ | Burkholder et al., 2015 |
| O($^1D_2$) + $O_2$ $\rightarrow$ O + $O_2$ $\rightarrow$ $O_3$ + $O_2$ | $3.93 \times 10^{-11}$ | Burkholder et al., 2015 |
| O($^1D_2$) + $N_2$ $\rightarrow$ O + $N_2$ $\rightarrow$ $O_3$ + $N_2$ | $3.05 \times 10^{-11}$ | Burkholder et al., 2015 |
| O($^1D_2$) + $H_2O$ $\rightarrow$ 2 OH | $1.97 \times 10^{-10}$ | Burkholder et al., 2015 |


Note:  Ground state hydrogen and oxygen atoms are assumed to instantaneously attach to $O_2$ under the photolysis conditions.  Photolysis of the $H_2O_2$ product at both 184.9 nm and 253.7 nm is an insignificant source of OH compared to the photolysis of water and reaction of O($^1D_2$) with water.  Photolysis of $O_3$ at 184.9 nm is only ~5% of that at 253.7 nm, and the quantum yield for O($^1D_2$) production is only about
50% of that at 253.7 nm and is ignored in the model.





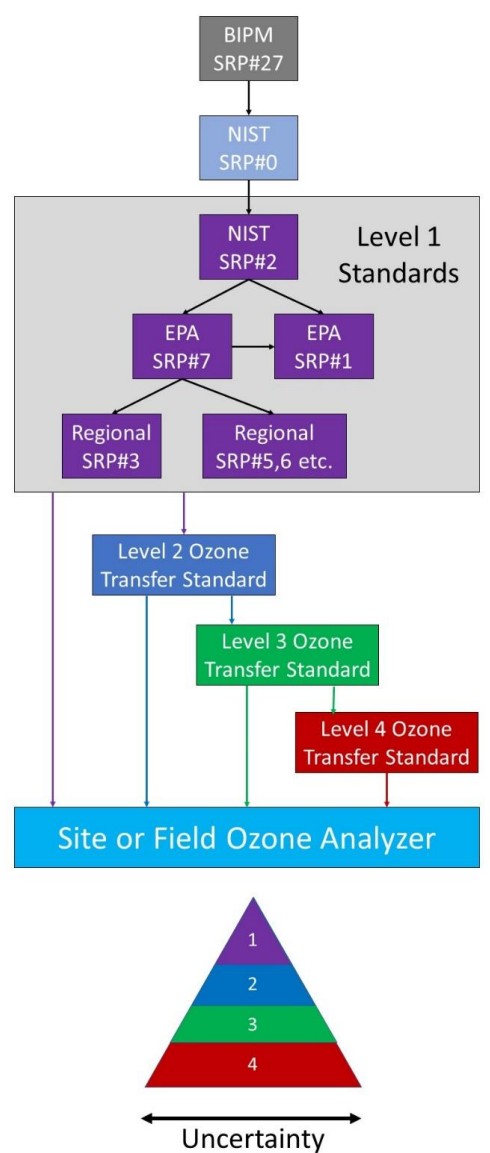


Figure 1.  U.S. EPA ozone transfer standard traceability.





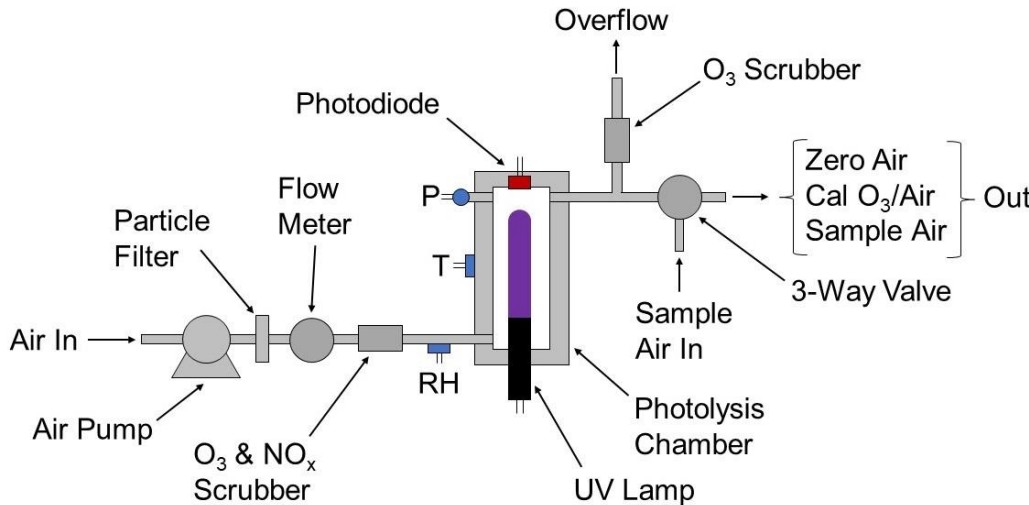


Figure 2. Schematic diagram of the 2B Technologies Model 306 Ozone Calibration Source.





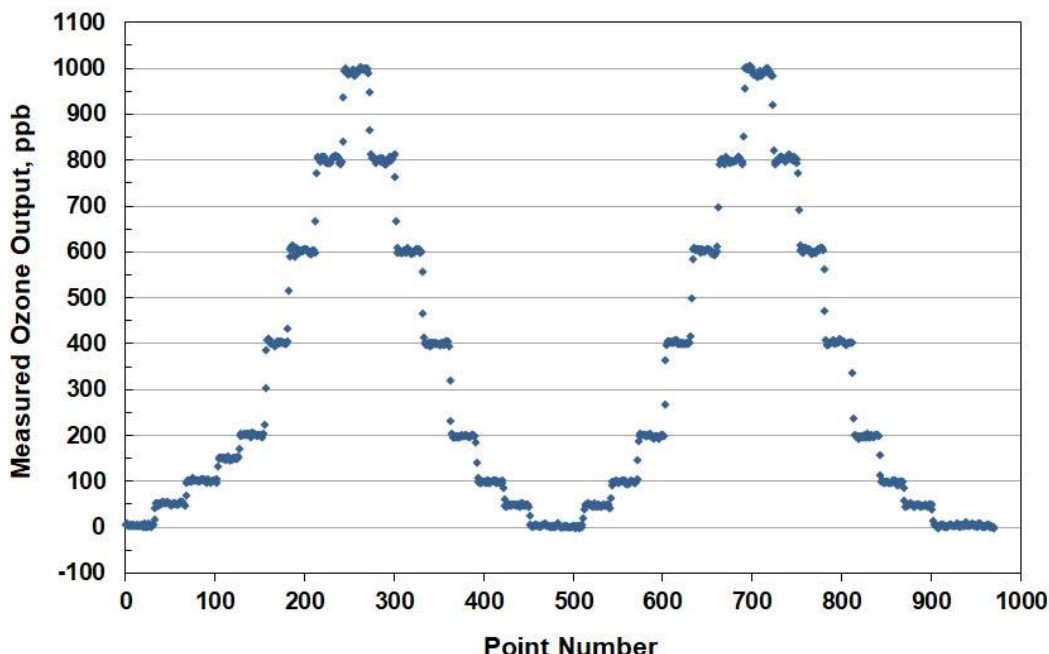

Figure 3. Measured Output of a Model 306 Ozone Calibration Source where the ozone mixing ratio was

500        systematically varied in steps of 50 and 200 ppb (30 points = 5 minutes), as described in the text.





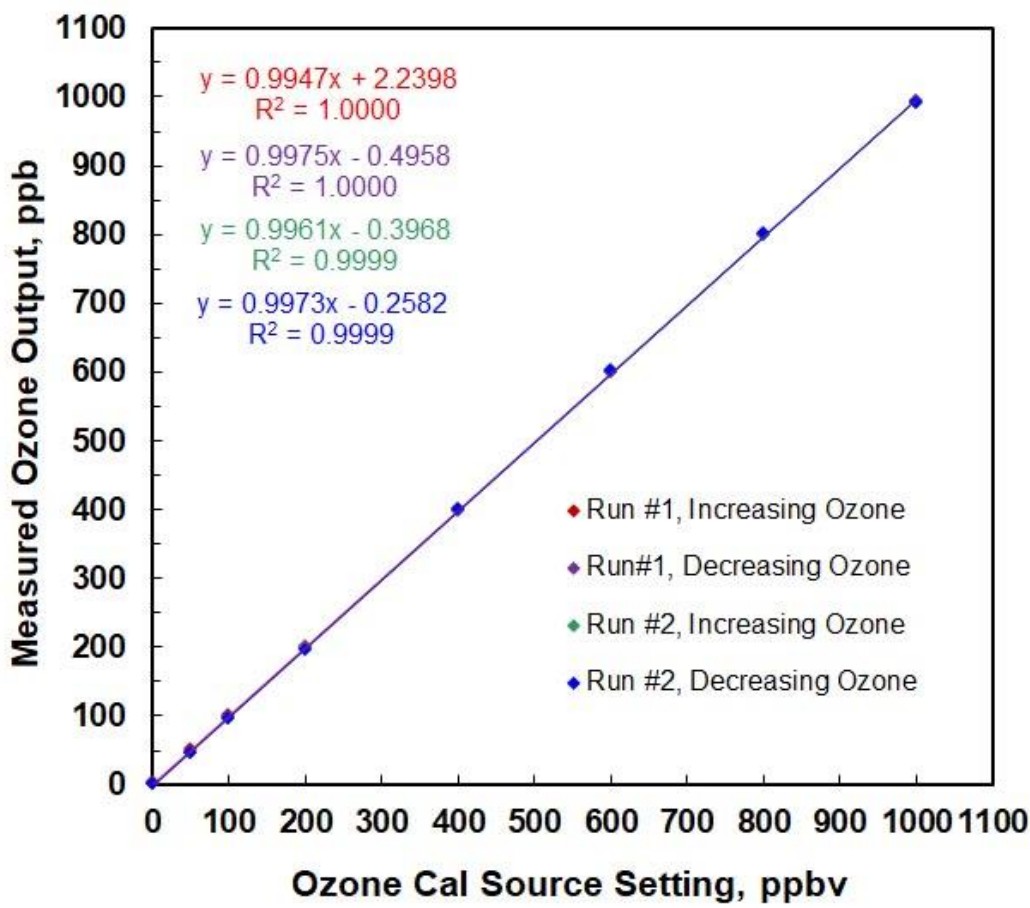

Figure 4. Linear regression for the measured outputs of a Model 306 Ozone Calibration Source of Fig. 3.

Note the excellent agreement among the four data sets of increasing and decreasing ozone output

505        concentration.  Note that the four regression lines are indistinguishable.


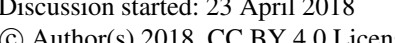


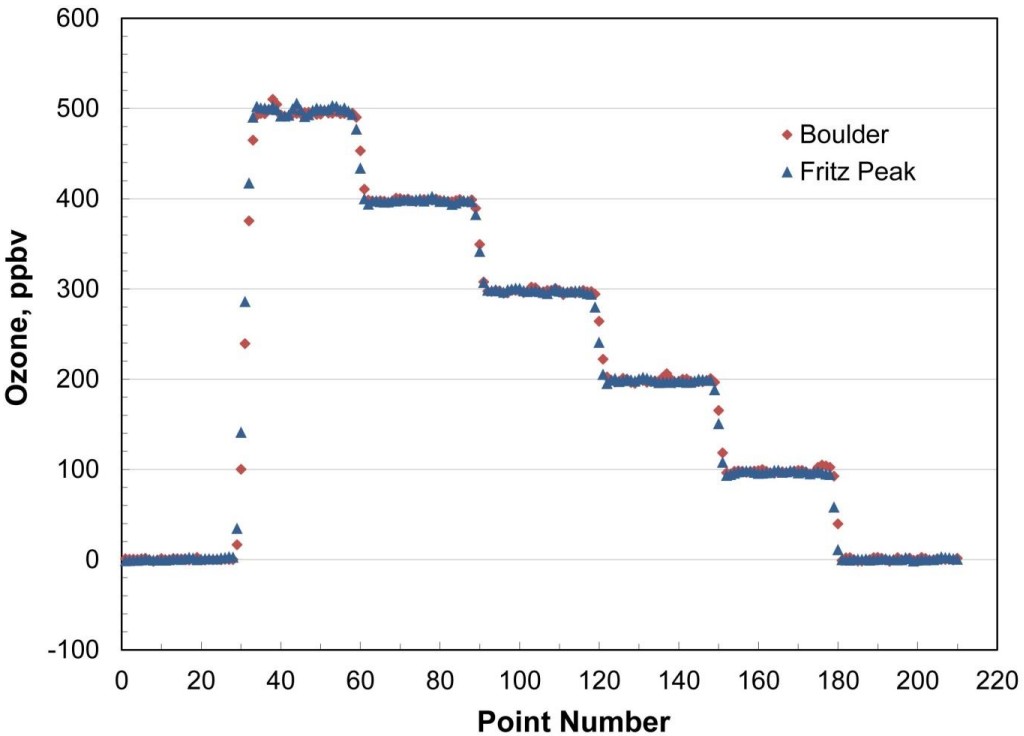

Figure 5. Comparison of ozone output mixing ratios in Boulder, Colorado (5430 ft, 1,655 m altitude) and
510       Fritz Peak (9020 ft, 2749 m altitude) as measured by a 2B Model 202 Ozone Monitor (30 points = 5
minutes).



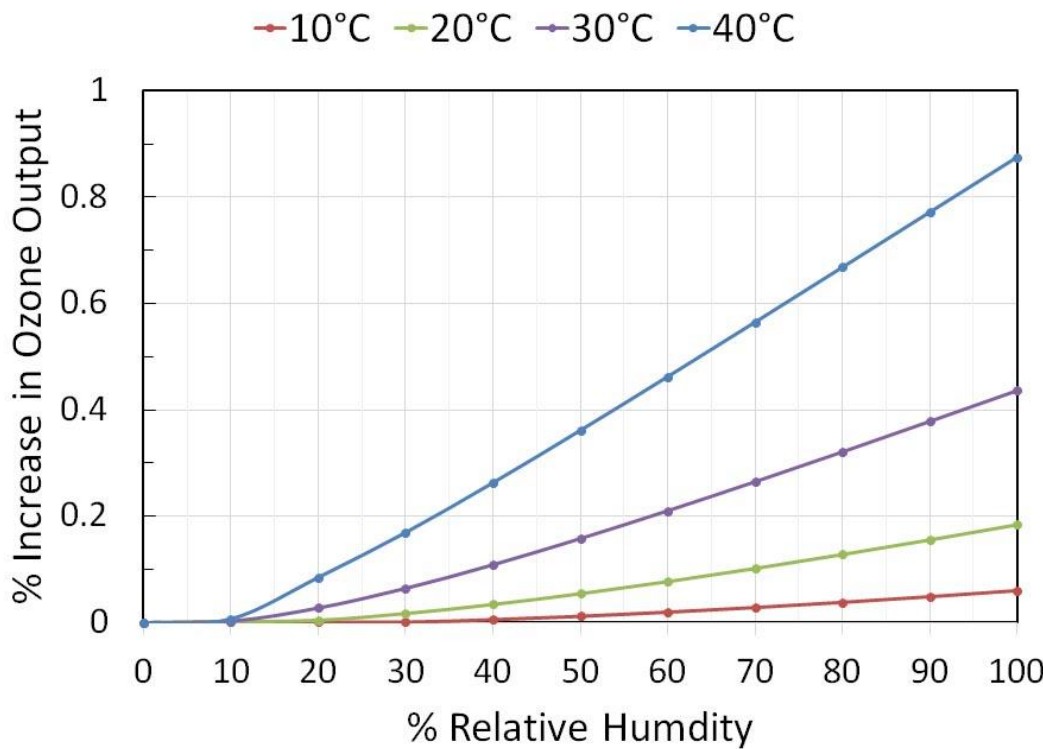

Figure 6.  Calculated percent increases in the ozone mixing ratio output (for a target of 100 ppb of $O_3$)

from the ozone calibration source due to photochemical reactions as a function of temperature

and relative humidity.



### (a) No Correction for Humidity

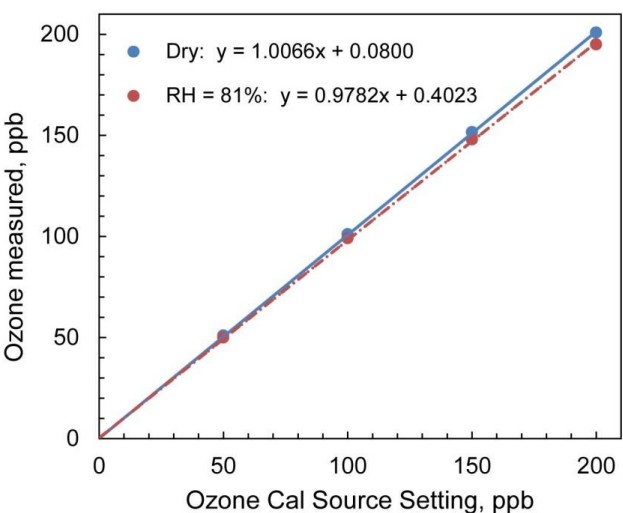

Dry: y = 1.0066x + 0.0800

RH = 81%: y = 0.9782x + 0.4023

### (b) Humidity Correction Applied

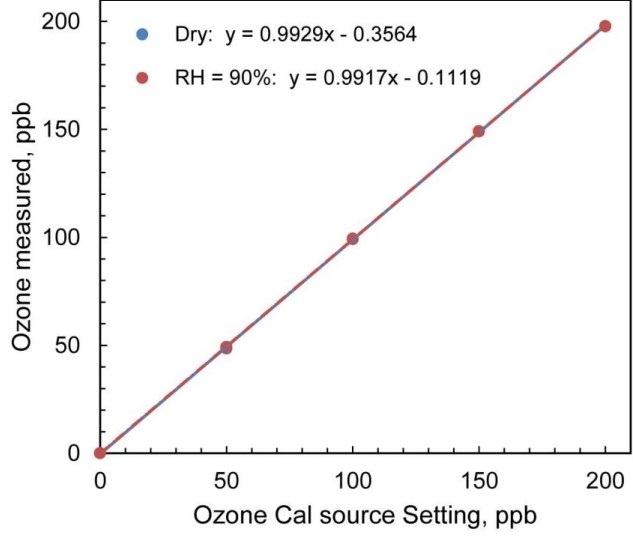

Dry: y = 0.9929x - 0.3564

RH = 90%: y = 0.9917x - 0.1119


Figure 7. Comparison of the output of the Model 306 ozone calibrator for dry and humid air for (a) no
firmware corrections for humidity and (b) firmware corrections applied based on in-line humidity
measurements.