# Peer review of "Portable Ozone Calibration Source Independent of Changes in Temperature, Pressure and Humidity for Research and Regulatory Applications"

_Atmospheric Measurement Techniques, 2018_

## Referee Comment (RC1) · Anonymous Referee #1 · 30 May 2018

General comments

Birks et al. present a technical paper describing a portable ozone calibrator developed by 2B Technologies. The paper is very well written and covers all the technical aspects for qualifying this equipment as an EPA Level 4 transfer. The interest of this equipment is to be easily portable, with low weight and low power consumption. The authors present clearly the technical layout and specifications of the instrument, with all figures being very clear and well presented. Based on theory and experimental tests, the authors prove that the instrument operation is free of pressure dependence

or water vapour interference. I recommend the publication of this paper with only minor corrections.

Specific comments

1. For "Air In", the instrument use a chemical scrubber (Line 150 and Figure 2) to remove O3 and NOx. The authors should specify the composition of this scrubber. A related question is the capacity of this scrubber to remove potential atmospheric interferences (as VOC) in highly polluted areas.

2. Table 2 should specify the robustness of the instrument to be used as EPA transfer (estimation of how often it must be calibrated through a higher level EPA transfer).

3. RH sensor HIH8000 is installed in the flow path upstream the cell. Its layout in the tubing should be slightly described. Its response time should be specified, to prove that potential rapid variations of RH will be included in the lamp intensity process for constant O3 production.

Technical corrections

Line 334: the mathematic formula should use slightly bigger characters police, in order to be more readable.

---

## Referee Comment (RC2) · Anonymous Referee #2 · 8 Jun 2018

The authors present a portable ozone calibration source that can serve as a transfer standard for the calibration of ozone monitors, e.g. those deployed in air quality monitoring stations (fulfils the requirements of a U.S. EPA level 4 transfer standard). The manuscript is very clear and well written and actually, it is complete and includes a thorough and interesting discussion about the effect of humidity on the generated O3 mixing ratio. In fact, I did not found any errors or things that should be changed or corrected. Nevertheless, I'm reluctant in recommending the manuscript for publication in AMT, because it is a description (although very detailed and correct) of a commer-

cial instrument. A good part of the information in the manuscript is already available on the 2B website within the product description of the Model 306 Ozone Calibration Source (https://www.twobtech.com/model-306-ozone-cal-source.html) and the operation manual that can be downloaded from the website (Figures 2, 3, 4 and Table 2 of the manuscript). I therefore think that the manuscript does not provide sufficient novel information to justify publication in a research journal like AMT. However, this is rather a political than a scientific or technical question and the decision should been taken by the Editor. Regarding content, the manuscript is fine.

---

## Referee Comment (RC3) · Anonymous Referee #3 · 10 Jun 2018

The manuscript describes a portable ozone calibration source to be easily used at field stations. The O3 production is based on oxygen photolysis at 184.9 nm using a mercury lamp. Stable O3 concentration are achieved by controlling the residence time in the photolysis chamber and by monitoring the lamp intensity at its emission wavelength at 253.7 nm. No zero air source is required and implications for the uncertainties are discussed. The authors have proven its capability to be suitable as an ozone calibration source under the tested conditions. I recommend publication after addressing following comments.

[Figure]

General Comments:

The manuscript is well written and describes the device's concept including uncertainty analysis and experimental verification. Critical for the produced O3 mixing ratio are i.e. the lamp intensity and the phototube which need further information. How stable is the phototube and the O3 calibration source over time? A discussion about the dependency of O3 production on the lamp line widths, characteristic for the lamp used, should be included in the manuscript. The O2 absorption around 184.9 nm has fine spectra and therefore the absorption is highly dependent on the line widths of the lamp (e.g. Lanzendorf et al., Geophysical Research Letters, Vol. 24, no. 23, p. 3037-3038, 1997; Hofzumahaus et al., Geophysical Research Letters, Vol. 24, no. 23, p. 3039-3040, 1997; Ceasy et al., Geophysical Research Letters, Vol. 27, no. 11, p. 1651-1654, 2000).

Specific Comments:

Page 5, line 150: As no zero air gas is used ambient air is scrubbed for O3, NO and NO2. Please provide information what kind of scrubber is used and what its efficiencies are for O3 and NOx. In city environment with more than hundred ppb NOx, residual NOx could have a non- negligible effect on the produced ozone. In VOC rich environments, such as forested regions, the produced ozone has the potential to react with remaining ambient VOCs. Additionally, absorption of the UV light by VOCs could occur. Please estimate the uncertainty for your produced ozone concentration due to VOC reactions and absorption.

Page 5, line 153: The performance of the phototube is one of the critical devices in the setup. Please state, what kind of phototube is used. What is its long term stability?

Page 6, Line 165: What is the accuracy of the regulated flow?

Page 6, line 167: Please clarify what is meant by scaling the voltage of the photodiode.

Page 7, line 202: The measured precision is a combination of the precision of the O3

calibration source and the O3 monitor. The authors state that the measured precision of the O3 monitor at zero ozone is 2.1 ppb and the regression of the combined precision of O3 monitor and O3 calibration source has an intercept of 1.8 ppb. However, it is unclear, how a constant offset in the precision can be attributed to the O3 monitor alone.

Page 8, Line 229: What is the uncertainty using this approximation?

Page 8, Line 236: The obtained O3 mixing ratio is a function of O3 production and O3 loss. In equation (5) the time dependent loss terms , e.g. O3 photolysis (184.9 nm, 253.7 nm), have to be considered.

Page 9, line 272: The authors have estimated the effect of water on the flow meter reading and its absorption for one special case to be 0.5%. Is this the maximum deviation which can occur?

Page 16, Table 2: The author stated a lower precision and accuracy than described in the paper to account for potential variability among individual instruments. How were these numbers derived? What is the reason for this variability?

Technical Comments:

Page 1, line 22: Not consistent: Later in the manuscript response time was stated to be < 30 s.

Page 3, Line 69: "Because ozone is an unstable gas, easily decomposing to molecular oxygen, calibrations ..." Please add that ozone is not stable in gas cylinders, e.g. "Because ozone is an unstable gas, easily decomposing to molecular oxygen in gas cylinders, calibrations ..."

Page 6, line 174: Please specify the type and material of the three-way solenoid valve.

Page 8, Line 223: "... and the oxygen concentration (cO2) in air at a temperature of 298 K ..." Please change to: ".. and the oxygen concentration (cO2) in dry air at a

temperature of 298 K ..."

Page 8, Line 223: Please use SI units.

Page 8, Line 245: Please add the range in which the flow can be maintained.

Page 8, line 250: Please quantify indistinguishable.

Page 11, line 321: Please quantify "sufficiently accurate".

Page 11, line 337: Please specify type and material of the three-way valve.

---

## Author Comment (AC1) · 2 Jul 2018

We would like to thank all three referees for their effort and their thoughtful comments. We have included our responses to all three reviewers in this supplement followed by an annotated version of the manuscript. Reviewers' comments are in *italics*, followed by our response to each comment. Changes to the manuscript are in red font within each response and in the annotated manuscript. Line and page numbers denoted our responses refer to the annotated manuscript included here.

**Anonymous Referee #1**

*General comments*
*Birks et al. present a technical paper describing a portable ozone calibrator developed by 2B Technologies. The paper is very well written and covers all the technical aspects for qualifying this equipment as an EPA Level 4 transfer. The interest of this equipment is to be easily portable, with low weight and low power consumption. The authors present clearly the technical layout and specifications of the instrument, with all figures being very clear and well presented. Based on theory and experimental tests, the authors prove that the instrument operation is free of pressure dependence or water vapour interference. I recommend the publication of this paper with only minor corrections.*

*Specific comments*
*1. For "Air In", the instrument use a chemical scrubber (Line 150 and Figure 2) to remove O3 and NOx. The authors should specify the composition of this scrubber. A related question is the capacity of this scrubber to remove potential atmospheric interferences (as VOC) in highly polluted areas.*

> We have clarified the details concerning the scrubber in response to both this comment and a similar comment by Referee #3. The scrubber is made of Carulite (a combination of copper and manganese oxides) which catalytically both destroys ozone and oxidizes NO to $NO_2$. We have changed the manuscript (page, 5, lines 152-154) to read:
> "…and a chemical scrubber to remove ozone and NO (which can react relatively rapidly with the ozone produced). The scrubber consists of Carulite, which catalytically destroys ozone and oxidizes NO to $NO_2$. $NO_2$ is not removed. Air then enters the photolysis chamber containing a low-pressure mercury lamp…". This scrubber is catalytic for both ozone destruction and NO oxidation, so that it has nearly limitless capacity. Further comments concerning possible VOC interferences are addressed in Specific Comment 1 from Referee #3.

*2. Table 2 should specify the robustness of the instrument to be used as EPA transfer (estimation of how often it must be calibrated through a higher level EPA transfer).*

> We have added an entry into Table 2 for the Recommended Calibration time along with an accompanying footnote description: "The recommended calibration time is the maximum time between validation of the Ozone Calibration Source with an independent EPA-certified standard."

*3. RH sensor HIH8000 is installed in the flow path upstream the cell. Its layout in the tubing should be slightly described. Its response time should be specified, to prove that potential rapid variations of RH will be included in the lamp intensity process for constant O3 production.*

We have changed the text (page 12, lines 357-361) to read: ,
"…a humidity sensor (Honeywell, HIH8000) was installed in the flow path via a tee with the sensor head protruding into the main flow immediately upstream of the photolysis cell…" to describe how the sensor was mounted within the instrument.  We have changed the next sentence to read: "The sensor provides simultaneous measurements of relative humidity (RH) and temperature with a response time of ~ 10 seconds so that mixing ratios…"
It should be noted that the inlet scrubber tends to act as a temporary reservoir for humidity, thus dampening any rapid changes in water vapor concentration.

*Technical corrections*
*Line 334: the mathematic formula should use slightly bigger characters police, in order to be more readable.*

We have increased the size of the equations in the text; however, we expect that this will be altered to comply with the journal standards if the manuscript is accepted for final publication.
* * *
**Anonymous Referee #2**

*The authors present a portable ozone calibration source that can serve as a transfer standard for the calibration of ozone monitors, e.g. those deployed in air quality monitoring stations (fulfils the requirements of a U.S. EPA level 4 transfer standard). The manuscript is very clear and well written and actually, it is complete and includes a thorough and interesting discussion about the effect of humidity on the generated O3 mixing ratio. In fact, I did not found any errors or things that should be changed or corrected. Nevertheless, I'm reluctant in recommending the manuscript for publication in AMT, because it is a description (although very detailed and correct) of a commercial instrument. A good part of the information in the manuscript is already available on the 2B website within the product description of the Model 306 Ozone Calibration Source (https://www.twobtech.com/model-306-ozone-cal-source.html) and the operation manual that can be downloaded from the website (Figures 2, 3, 4 and Table 2 of the manuscript). I therefore think that the manuscript does not provide sufficient novel information to justify publication in a research journal like AMT. However, this is rather a political than a scientific or technical question and the decision should been taken by the Editor. Regarding content, the manuscript is fine.*

We respect Referee #2 opinion and certainly realize that some of the data presented here is also contained on our website and manuals; however we do feel that the manuscript goes into greater depth on the basic chemistry involved in our Ozone Calibrator to show why it is pressure independent and has only a very small (and correctable) humidity dependence. We also feel that the manuscript provides the general scientific community a good overview into how ozone measurements are validated across large monitoring networks and what is required for an instrument or calibration unit to be certified within these networks.

———————————————————

**Anonymous Referee #3**

*The manuscript describes a portable ozone calibration source to be easily used at field stations. The O3 production is based on oxygen photolysis at 184.9 nm using a mercury lamp. Stable O3 concentration are achieved by controlling the residence time in the photolysis chamber and by monitoring the lamp intensity at its emission wavelength at 253.7 nm. No zero air source is required and implications for the uncertainties are discussed. The authors have proven its capability to be suitable as an ozone calibration source under the tested conditions. I recommend publication after addressing following comments.*

*General Comments:*

*The manuscript is well written and describes the device's concept including uncertainty analysis and experimental verification. Critical for the produced O3 mixing ratio are i.e. the lamp intensity and the phototube which need further information. How stable is the phototube and the O3 calibration source over time? A discussion about the dependency of O3 production on the lamp line widths, characteristic for the lamp used, should be included in the manuscript. The O2 absorption around 184.9 nm has fine spectra and therefore the absorption is highly dependent on the line widths of the lamp (e.g. Lanzendorf et al., Geophysical Research Letters, Vol. 24, no. 23, p. 3037-3038, 1997; Hofzumahaus et al., Geophysical Research Letters, Vol. 24, no. 23, p. 3039-3040, 1997; Ceasy et al., Geophysical Research Letters, Vol. 27, no. 11, p. 1651-1654, 2000).*

The stability of the detector is addressed in Specific Comment #3 below. Referee #3 is correct that the spectral overlap between the lamp emission lines and the fine structure of the $O_2$ Schumann-Runge bands is critical to the linearity and stability of the Ozone Calibrator. It is essential to maintain a constant lamp/photolysis cell temperature (40 ± 1 °C) since both the $O_2$ absorption cross section and line broadening of the Hg emission lines (primarily Doppler broadening) are sensitive to temperature. It is of key importance that the lamp temperature not change appreciably as its intensity is changed. If this occurred to a significant extent, it would result in a nonlinear ozone output due to a changing spectral overlap. Since the measured output is quite linear with lamp intensity, this suggests minimal change in the line broadening of the 184.9 nm Hg line in our Calibrator. We have included the following discussion concerning this on page 7 at line 205: Past work has shown that the "effective" absorption cross section of $O_2$ using a Hg lamp at 184.9 nm varies with $O_2$ concentration (Creasey et al., 2000; Cantrell et al., 1997). This has been shown to be due to poor overlap between the Hg lamp emission lines and the highly structured $O_2$ absorption in the Schumann-Runge bands (Lanzendorf et al., 1997). Both the $O_2$ absorption lines and the broadening of the Hg emission lines are sensitive to temperature and, therefore, controlling the photolysis cell temperature at 40 °C (± 1 °C) is critical to maintaining constant spectral overlap. A changing spectral overlap could result from self-heating within the Hg lamp as the intensity is increased and result in nonlinear ozone production. However, the high degree of linearity observed (Fig. 4) suggests that the lamp temperature (thus the spectral overlap) remains constant over the range of lamp intensities employed.

A varying spectral overlap may also play a role in the long-term stability as the lamp emission degrades over time. However we have found that the long-term calibration appears to be more sensitive to contamination of the windows which slowly attenuates the 253.7 nm light used to control the lamp intensity. That is why we recommend validating the output of the Ozone Calibrator at least annually with an independent standard and we have included this recommendation in Table 2 at the suggestion of Referee #1.

*Specific Comments:*
*1. Page 5, line 150: As no zero air gas is used ambient air is scrubbed for O3, NO and*

*NO2. Please provide information what kind of scrubber is used and what its efficiencies are for O3 and NOx. In city environment with more than hundred ppb NOx, residual NOx could have a non- negligible effect on the produced ozone. In VOC rich environments, such as forested regions, the produced ozone has the potential to react with remaining ambient VOCs. Additionally, absorption of the UV light by VOCs could occur. Please estimate the uncertainty for your produced ozone concentration due to VOC reactions and absorption.*

We have clarified the details concerning the scrubber in response to both this comment and a similar comment by Referee #1. The scrubber is made of Carulite (a combination of copper and manganese oxides) which catalytically both destroys ozone and oxidizes NO to $NO_2$. We have changed the manuscript (page, 5, lines 152-154) to read:

"…and a chemical scrubber to remove ozone and NO (which can react relatively rapidly with the ozone produced). The scrubber consists of Carulite, which catalytically destroys ozone and oxidizes NO to $NO_2$. $NO_2$ is not removed. Air then enters the photolysis chamber containing a low-pressure mercury lamp…".

We also agree with Referee #3 that trace gases (primarily $NO_2$ and VOCs) could potentially interfere with ozone generated by either chemical reactions or light absorption. To address this we have changed the subtitle of Section 3.3 (beginning on page 9, line 272) to: "**Effect of Trace Gases and Water Vapor on the Ozone Output Mixing Ratio**" and have added the following two paragraphs:

Trace gases that are not removed by the inlet scrubber can affect the ozone output in two ways: (1) direct chemical reaction with the ozone produced or (2) by light absorption that can affect either the overall light intensity (reducing $O_2$ photolysis) or producing reactive photoproducts. $NO_2$ and volatile organic compounds (VOCs) are of primary concern (water vapor is a special case and considered separately below). Chemical loss of ozone in the photolysis cell is limited by the short residence time ($\tau_{res}$ ~ 0.06 s); however, one must also consider the transit time to an analyzer which is to be calibrated. For a typical transit time of ~ 1 s (1 m length of 4 mm i.d. tubing and an analyzer flow rate of 1 L min$^{-1}$) and assuming an $NO_2$ or VOC concentration of 500 ppb (extremely polluted urban area), a rate coefficient of > 1 x 10$^{-15}$ cm$^3$ molec$^{-1}$ s$^{-1}$ is required to remove 1% of the ozone produced. Rate coefficients for $NO_2$ and relatively stable VOCs (atmospheric lifetime > 3 hr) with ozone are typically more than an order of magnitude smaller (Burkholder et al.,

2015, Finlayson-Pitts and Pitts, 2000). There are VOCs that are much more reactive with ozone (most notably terpenoid compounds in forested areas), but due to this high reactivity, their ambient concentrations are rarely above 1 or 2 ppb (e.g., Yee et al., 2018). Furthermore, these reactive VOCs have been shown to be effectively removed by $MnO_2$-type scrubbers (Pollmann et al., 2005).

Photolysis of $NO_2$ and possible VOCs cannot compete with $O_2$ photolysis due to overwhelming concentration difference. Even though aromatic VOCs typically have large absorption cross sections at 184.9 nm (~ $10^{-16}$ $cm^2$ $molec^{-1}$, Keller-Rudek et al., 2013), a mixing ratio of 200 ppb results in a VOC photolysis rate (= $I\sigma_{VOC}[VOC]$) that is only 1% of the $O_2$ photolysis rate (Eq (4)). Therefore, the presence of trace VOCs and $NO_2$ are not large enough to either affect the light intensity or generate substantial amounts of photoproducts that could impact the ozone concentration produced.

2. *Page 5, line 153: The performance of the phototube is one of the critical devices in the*

*setup. Please state, what kind of phototube is used. What is its long term stability?*

The detector is a solid state silicon photodiode. This is already described in the text and we have included the make/model (Hamamatsu, S12742-254) in the existing description on page 4, line 156.

We have also included a sentence and reference about the long-term stability of silicon photodiodes at page 6, line 158: "Solid-state silicon photodiodes are known to maintain their original sensitivity longer than any other photodetector and, as such, are used as NIST transfer standards (Ryer, 1998). This translates to long-term stability in the ozone output of the Ozone Calibrator."

3. *Page 6, Line 165: What is the accuracy of the regulated flow?*

The mass flow meter is the Model 4121 made by TSI. It has a stated accuracy of ± 2 % which is verified by comparison with a NIST traceable flow standard (Bios Defender Model 530) in our laboratory. In line 168 (page 6), where we give the make/model number of the flow meter, we have added the stated accuracy.

4. *Page 6, line 167: Please clarify what is meant by scaling the voltage of the photodiode.*

We apologize for the uncertainty, the phrase "voltage of the photodiode" is unclear. We have changed this to read (page 6, line 173): "In addition to controlling the volumetric flow rate the target photodiode signal (corresponding to the target output ozone) is scaled to the instantaneously measured volumetric flow rate in order to compensate for flow rate fluctuations, (e.g., higher flow rates require higher target photodiode signals). …"

5. *Page 7, line 202: The measured precision is a combination of the precision of the O3*

*calibration source and the O3 monitor. The authors state that the measured precision of the O3 monitor at zero ozone is 2.1 ppb and the regression of the combined precision of O3 monitor and O3 calibration source has an intercept of 1.8 ppb. However, it is*

*unclear, how a constant offset in the precision can be attributed to the O3 monitor alone.*

At an ozone mixing ratio of zero – there is no contribution to the precision from the $O_3$ calibration source (the lamp is turned off). Thus the $O_3$ monitor is responsible for the total uncertainty in this situation. However, Referee #3 is correct that the increase in precision with $O_3$ mixing ratio is a combination of both the calibrator and $O_3$ monitor and cannot be separated. We only noted the intercept in the plot of precision vs. [$O_3$] to show that it was nearly the same as that measured in the absence of ozone (1.8 ppb vs. 2.1 ppb). Assuming that the increase in precision was due solely to the Calibrator output gives the estimated 0.4% precision reported; however this is actually more of an upper limit. We have changed the text at page 7, line 218 to read: "Thus, assuming this increase is due solely to the Ozone Calibrator (and not the Model 202 monitor), the precision of the ozone output is about 0.4% of the target concentration (e.g., ±0.4 ppb at 100 ppb $O_3$ and ±4 ppb at 1,000 ppb $O_3$)."

*6. Page 8, Line 229: What is the uncertainty using this approximation?*

The calculation in the preceding two sentences (page 8, lines 241-247) indicates that assuming optically thin conditions only results in about a 1.2% attenuation of the light. This was erroneously reported at 0.13% and we have corrected this. Therefore, the approximation in Eq. (4) is good to within 1.2%. We have also corrected the text to reflect that the photolysis occurs at 40°C (and not 298 K as was previously written).

*7. Page 8, Line 236: The obtained O3 mixing ratio is a function of O3 production and O3*

*loss. In equation (5) the time dependent loss terms , e.g. O3 photolysis (184.9 nm, 253.7 nm), have to be considered.*

Ozone photolysis is unimportant since it produces oxygen atoms (either directly or via quenching of excited state $O(^1D)$ by nitrogen and oxygen) that primarily recombine with $O_2$ reforming ozone. The only loss terms that can be important involve reactions of $O(^1D)$ atoms (from ozone photolysis) that are not quenched to ground state oxygen atoms ($O(^3P)$). This primarily will occur due to the presence of water vapor and is discussed in Section 3.3 and shown to have a minimal effect on the ozone produced.

*8. Page 9, line 272: The authors have estimated the effect of water on the flow meter*

*reading and its absorption for one special case to be 0.5%. Is this the maximum deviation which can occur?*

Yes – the case examined pertains to conditions of 100% RH at 40°C (described in the preceding paragraph – page 10, lines 298-300), which is a likely maximum water vapor concentration in ambient air.

*9. Page 16, Table 2: The author stated a lower precision and accuracy than described in*

*the paper to account for potential variability among individual instruments. How were these numbers derived? What is the reason for this variability?*

As noted in the comment above – it is nearly impossible to separate the precision and accuracy of the Ozone Calibrator to that of the ozone monitor that is used for standardization. The specifications given in Table 2 are chosen to encompass observed uncertainties observed in our typical ozone monitors.

*Technical Comments:*

1. *Page 1, line 22: Not consistent: Later in the manuscript response time was stated to*

*be < 30 s.*

We have changed to text to read "30 sec" – consistent with later mentions.

2. *Page 3, Line 69: "Because ozone is an unstable gas, easily decomposing to molecular*

*oxygen, calibrations ..." Please add that ozone is not stable in gas cylinders, e.g.*
*"Because ozone is an unstable gas, easily decomposing to molecular oxygen in gas*
*cylinders, calibrations ..."*

We have changed the text as suggested.

3. *Page 6, line 174: Please specify the type and material of the three-way solenoid valve.*

The valve is from Parker Hannifin and is actually a custom part number – we have included the description: "A three-way solenoid valve (Parker-Hannifin, nickel plated V2 miniature valve) is installed…"

4. *Page 8, Line 223: "... and the oxygen concentration (cO2) in air at a temperature of*

*298 K ..." Please change to: ".. and the oxygen concentration (cO2) in dry air at a*
*temperature of 298 K ..."*

We have changed the text as suggested.

5. *Page 8, Line 223: Please use SI units.*

We have included the pressure in kPa as well as atm.

6. *Page 8, Line 245: Please add the range in which the flow can be maintained.*

The flow is maintained to within 1% (30 $cm^3$ $min^{-1}$ for a 3.0 L $min^{-1}$ flow rate). We have included this in the initial description of the instrument on page 6, line 171: "…and regulates the volumetric flow rate to be 3.0 L $min^{-1}$ (± 1%) by means of pulse-width modulation of the power supplied to the pump."

7. *Page 8, line 250: Please quantify indistinguishable.*

We have changed the text to read: "The output ozone mixing ratios are at these two altitudes are indistinguishable (within 2%), as predicted by theory."

8. *Page 11, line 321: Please quantify "sufficiently accurate".*

Monteith and Unsworth (2008) state that saturation vapor pressures calculated via Eq (16) are within "1 Pa of the exact values" up to 35°C. Note that the humidity measurement is made at near-ambient temperature and prior to the gas entering the heated photolysis cell. We have added this accuracy quote in the text on page 12, line 364.

9. *Page 11, line 337: Please specify type and material of the three-way valve.*

We have included the manufacturer and product number of the valve. The valve was made of stainless steel, but the material choice for this valve is less critical because this three-way valve only contacts dry zero-grade air in the experiment described.

[revised manuscript text omitted]

---

## Author Response (AR2)

**Author's Responses to Associate Editor comments:**
*Below are our itemized responses to the Associate Editor's comments along with an annotated version of the manuscripts showing our changes. We thank the editor for his time and believe that his comments will add to the quality of our manuscript. Our responses are in italics below and changes to the manuscript are in red font.*
**Associate Editor Decision**: Publish subject to minor revisions (review by editor)
 (18 Jul 2018) by Andreas Hofzumahaus

Comments to the Author:
The paper has been evaluated by three reviewers. All referees rate the importance and quality of the paper as good, with the exception of referee #3 who considers the scientific quality to be only fair. Referees #1 and #3 recommend publication subject to minor revisions, whereas referee #2 recommends to reject the paper because of overlap with technical documentation from the manufacturer. In their reports, Referee #1 and #3 have raised a number of scientific and technical questions about the instrument, which have been adequately answered in the author's reply. Corresponding information and improvements have been added by the authors to the manuscript. After revision, the manuscript contains considerably more scientific information on the functioning and quality of the ozone generator described than can be found in the technical documentation available on the manufacturer's website. Therefore, I agree with referees #1 and #3 that the paper after revision merits publication in AMT.
I have a few minor points that should be changed before the paper can be published.

1. Lines 206 - 208: Cantrell et al. (1997) have investigated the absorption cross section of water vapor, but not of O2. Besides Creasey et al. (2000), other appropriate references are Hofzumahaus et al. (1997) and Lantzendorf et al. (1997) as pointed out by referee #3.

*We have included the Lanzendorf et al (1997) and Hofzumahaus et al. (1997) references in the statement on lines 206-208 for completeness (we also rearranged the order to reflect chronological order). We have not removed the Cantrell et al (1997) reference. Although the Cantrell et al (1997) study did not report O2 cross sections, they were the first to report that the O2 cross section varied with O2 pressure when measured using a Hg pen ray lamp source (as cited by Lanzendorf et al., 1997).*

2. Line 153: Carulite is a registered trademark for the catalyst. The material (a combination of copper and manganese oxides) should be explicitly mentioned in the paper (as in the reply to referee #3).

*We have included the chemical composition of Carulite®. We kept the registered trademark name as well since it provides a means for researchers to easily search for the material if interested.*

3. The ozone generator can provide accurate ozone mixing ratios at reduced atmospheric pressure as is shown by tests at different altitudes (Figure 5). Can the instrument also provide calibration gas at higher pressure (> 1atm), which could be useful in laboratory applications, and what would be the maximum useful outlet pressure?

*We have only tested the O3 calibrator up to 1 atm. Theoretically, the calibrator should work at higher pressures and would be limited by components such as the pressure sensor and the capacity of the pump to provide the required flow rate. Currently the pressure sensor is the limiting factor (range of 15 to 115 kPa). We have added the sentence at line 271: "
[revised manuscript text omitted]